# Assessing the Role of Lipopolysaccharide (LPS) Receptor (CD14) in Septic Cardiomyopathy: The Value of Immunohistochemical Diagnostics

**DOI:** 10.3390/diagnostics12040781

**Published:** 2022-03-23

**Authors:** Peter Michael Reil, Teodor Traian Maghiar, Narcis Vîlceanu, Andrei Pascalau, Claudia Teodora Judea Pusta, Florin Marcu, Simona Cavalu, Ovidiu Pop

**Affiliations:** 1Klinikum Ingolstadt GmbH, 85049 Ingolstad, Germany; info@klinikum-ingolstadt.de; 2Department of Morphology Sciences, University of Oradea, 410087 Oradea, Romania; mfmihai27@yahoo.com (T.T.M.); forensicanv@yahoo.com (N.V.); fmf@urodea.ro (C.T.J.P.); drovipop@yahoo.com (O.P.); 3Department of Psycho-Neuro-Sciences and Recovery, University of Oradea, 410087 Oradea, Romania; fmf@uroadea.ro; 4Faculty of Medicine and Pharmacy, University of Oradea, 410087 Oradea, Romania; scavalu@uoradea.ro

**Keywords:** septic cardiomyopathy, apoptosis, CD14, lipopolysaccharide, cytokines

## Abstract

Sepsis-induced myocardial dysfunction (SIMD) is one of the major predictors of morbidity and mortality of sepsis. A high percentage of patients with SIMD develop a status similar to cardiogenic shock. A high level of bacterial lipopolysaccharide (LPS) associated with an overexpression of CD14 acts as the trigger for the release of a broad spectrum of cytokines. Our study aimed to understand the correlation between septic cardiomyopathy and CD14 immunohistochemical expression. The study included 29 patients who died of septic shock. Increased values of membranous CD14 and soluble CD14 in the heart tissue were correlated with adverse patient evolution. A high cellular expression of CD14 was noted in the study group vs. the control group (*p* = 0.0013). Therefore, a close positive association between the amount of LPS related to sCD14 and the cellular expression of mCD14 is probable. By extrapolation, we suggest that a large amount of sCD14 detected in the cardiac tissue will activate the mCD14–TRL4–LBP–LPS complex, which in turn will induce an inadequate immune response, resulting in heart damage proportional to the amount of LPS. CD14 could represent a valuable marker for septic cardiomyopathy; thus, apoptosis of cardiomyocytes could be foreseen by its high value.

## 1. Introduction

Sepsis is a frequent cause of severe disease and death globally, being a major public health concern [1,2,3,4]. Sepsis-induced myocardial dysfunction was observed and described for the first time in 1984. Patients suffering from septic shock showed impaired cardiac systolic function by decreased left-ventricular ejection fraction (LVEF) [5,6]. Similar courses of reversible myocardial dysfunction due to the systemic inflammatory response syndrome were also shown in other critical illnesses [7,8]. Septic patients developing myocardial dysfunction have a significantly higher mortality (70%) [9]. The prevalence of the disease in septic patients ranges from 10% to 70%. This wide range is the result of a lack of formal diagnostic criteria and under-recognition of septic cardiomyopathy [10]. For a better understanding of the correlation between systemic inflammatory response and reversible myocardial dysfunction in sepsis, the levels of BCL2 protein, oncogene protein p53, and CD14 surface protein in patients who died from septic shock were investigated [6].

The human gene CD14 (cluster of differentiation 14) expresses a glycoprotein which is a component of the innate immune system, and it was the first described pattern recognition receptor. The membranous form (mCD14) is anchored mostly to the membrane of macrophages, but also that of neutrophils and dendritic cells. The soluble form (sCD14) appears after shedding from the membrane or as a result of direct secretion from intracellular vesicles [11], especially from the liver and monocytes. Both mCD14 and sCD14 are present in enterocytes [12]. Along with TLR4 (Toll-like receptor 4) and MD-2 (lymphocyte antigen 96), CD14 is a coreceptor for the detection of bacterial lipopolysaccharide (LPS) [13,14]. In the presence of lipopolysaccharide-binding protein (LBP), it binds to LPS. Additionally, it is able to recognize other pathogen-associated molecular patterns such as lipoteichoic acid [15].

Basically, the Toll-like Receptor (TLR) family recognizes several microbial products, including bacterial cell wall components and DNA [16]. Both preclinical studies (animal models) and clinical ones involving human subjects have proven an association between TLR4 mutations and a diminished degree of responsiveness to LPS [17]. Myocardial dysfunction after endotoxin exposure depends on the presence of cell-wall receptors such as TLR4 and CD14 [18,19]. It was demonstrated that CD14-deprived mice showed a normal cardiac function, being protected by LPS-induced shock [19], while others showed a decreased ventricular fraction [20]. However, the exact mechanism of action remains unclear. It can be assumed that cardio-depression may be induced by secondary cytokines such as TNF-α or endotoxins, as a result of CD14-mediated pathways [21,22].

In this context, the aim of this our study was to determine any significance of the mCD14 and sCD14 levels in the septic cardiomyopathy, and to evaluate the correlation with lipopolysaccharide-binding protein (LBP) by means of immuno-histological examinations.

## 2. Materials and Methods

### 2.1. Case Selection

The study included heart specimens collected from 29 consecutive cases of adults who died of septic shock. All cases were analyzed for membranous and soluble CD14 expression on the myocardial tissues. The exclusion criteria were based on the diagnosis of heart failure and malignant diseases. The control group consisted of 10 consecutive newborns who died in the same period, without any cardiac or malignant diseases. All cases were collected and documented from the Clinical County and Clinical City Hospital of Oradea (both located in Oradea, Romania), during the period January 2018–December 2018. From a histopathological point of view, all cases were processed, assessed, and diagnosed in the Pathology Department of Resident Laboratory Oradea, Romania.

### 2.2. Specimen Preparation and Immunohistochemical Investigation

Paraffin-embedded fixed cardiac tissues, 4 µm thick sections, were stained by Ventana Benchmark GX (Ventana Medical Systems Inc., Tucson, AZ, USA). Following the manufacturer’s instructions, the slides were deparaffinized using EZ prep solution (Ventana Medical Systems, Inc.), incubated with monoclonal antibodies, developed using the Opti View DAB detection kit (Ventana Medical Systems, Inc.), and counterstained with hematoxylin and bluing. For CD14 reactivity, sections were incubated with anti-CD14 primary monoclonal antibody (EPR1653, rabbit, IgG, membranous/cytoplasmatic, IVD-Ventana Medical Systems Inc., Tucson, AZ, USA) in accordance with the manufacturer’s protocol. For each run, a positive control slide (tonsil) was performed. The image acquisition was performed using a microscope with intelligent automation, Leica DM3000 LED and LAS EZ Software (provided by Leica Biosystem, Buffalo Grove, IL, USA) [6].

The interpretation was performed using the hotspot technique. The hotspot areas refer to areas with high expression. The patients included in the study were divided into two groups: <10% (low expression) and >10% (high expression) [23]. For data storage and statistical calculations, the statistical software MedCalc^®^ version 12.5.0.0 (MedCalc^®^ Software, Mariakerke, Belgium) and Microsoft^®^ Excel^®^ 2010 (Microsoft^®^ Corporation, Redmond, WA, USA) were employed. The interpretation of the results was conducted by assessing the probability of the null hypothesis (*p*), with *p* < 0.05 indicating a statistically significant difference between study groups. The study was conducted according to the guidelines of the Declaration of Helsinki and approved by the Ethics Committee of Resident Laboratory (protocol code 06/31 March 2018).

## 3. Results

Septic cardiomyopathy is a common feature of severe sepsis syndromes and results in impaired intrinsic cardiac contractility. According to the particular infection site, the distribution of the study group is presented in Table 1, while the epidemiological data including the age, gender, and social provenance are displayed in Table 2. The mean value of the age for the study group was 60.3 years (±16.5). The mean value of the age (in months) for the control group was 4.2 months (±2.6). The myocardial cell expression of CD14 protein was significantly increased in the study group, as compared with control cases, where the expression rate remained under 10% (*p* = 0.0013) (Table 3). The patients with a higher expression of CD14 were older, but the difference did not reach the limit of statistical significance (*p* = 0.1204) (Table 4). There was a positive correlation between cellular (membranous, mCD14) and intravascular (soluble, sCD14) CD14 values (*p* = 0.0077) (Table 5). Females showed a higher incidence of increased intravascular CD14, but this difference did not reach the statistically significant threshold (Table 6). There was no difference in sCD14 expression rates related to the provenance (*p* = 0.9271), infection site (*p* = 0.7301), and age (*p* = 0.4976, with a tendency toward older age (Table 7). 

Evaluation of microscopic sections using only conventional H&E staining does not reveal the presence of LPS. H&E staining emphasizes interstitial edema, muscle fiber suffering, myocardial cytolysis, and the presence of inflammatory cells (Figure 1A). On the other hand, the use of auxiliary immunohistochemistry techniques provides valuable, additional information. The microscopic image shows myocardial cells, among which numerous inflammatory cells with a membranous, brown color indicate immune expression of CD14 as a cluster of differentiation. mCD14 is a multifunctional glycoprotein expressed mainly on the membrane surface of macrophages. According to Figure 1B, CD14 is also highly distributed on the cell surface of neutrophils (mCD14). Analysis of blood vessels revealed how inflammatory cells such as neutrophils/monocytes (mCD14) migrate into the myocardium through the vascular wall. The soluble part sCD14 (LPS) remains strictly confined in the blood vessel lumen (Figure 1C). The blood vessels are located intramuscularly, as mCD14 can also be seen with sCD14. In the lumen of this vessel, several granular extracellular brown areas (soluble CD14), as well as monocytes and neutrophils (mCD14), are noticeable (Figure 1D). 

## 4. Discussion

CD14 acts as a coreceptor for TLR4 and interferes with the LPS response. CD14 forms a complex with LPS and the LPS-binding protein (LBP), and this complex binds to the TLR4, inducing NF-κB-associated immune responses [24]. The onset of the immune responses induces the release of tumor necrosis factor alpha (TNF-α), IL-1, IL-6, and IL-8 [25]. Many studies concluded that a higher level of LPS associated with an overexpression of CD14 acts as a trigger for the release of a broad spectrum of cytokines. A higher cytokines level indicates a higher probability of an adverse immune response.

To the best of our knowledge, this is the first paper in which the expression of CD14 surface protein was evaluated in septic shock and septic cardiomyopathy in humans. Our study showed a positive association between markedly increased values of CD14 and an adverse patient evolution. A statistically significant increase in the level of membranous and soluble CD14 surface protein was found in the study group. Overall, there was a tendency for higher values in relation to older patients, but without statistical significance. The increasing values with age (despite not statistically significant) might be a clue for the elevated mortality level in older patients. There was no statistically relevant difference regarding the patient’s gender (*p* = 0.3990), provenance (*p* = 0.9892), or infection site (*p* = 0.3803). The results of our study regarding patient gender with septic shock are in contradiction with other studies [26]. One explanation could be the relatively small number of studies worldwide. Most studies revealed that female sex hormones exhibit protective effects in septic shock. Estrogens and their precursors might attenuate the levels of proinflammatory agents and reduce the effects for septic patients [26,27].

It is very well known that there are two types of LPS coreceptors: membrane-bound CD14 (mCD14) complexed with TLR4 and the LBP–LPS complex, and soluble CD14 (sCD14) complexed with LBP–LPS located in the blood. The expression of mCD14 on many epithelial cells such as human endometrial stromal cells may explain the cytokine storm [28]. The soluble form of CD14 is able to bind LPS and activate cells that otherwise do not express membrane-bound CD14 (e.g., endothelial, epithelial, and smooth muscle cells) [29,30]. The role of sCD14 was revealed by previous studies, suggesting that the CD14–TLR4–MD2 complex is involved in the phagocytosis of Gram-negative bacteria and intracellular signaling by LPS [31,32,33].

Our study revealed a positive association between cellular expression (membranous, mCD14) and intravascular (soluble, sCD14) levels. Despite the fact that the patient’s gender did not significantly influence the intravascular CD14 expression rate in the study group (*p* = 0.4612), the females showed a higher incidence. CD14 plays a double role: (1) as a component of the innate immune system, a pattern recognition receptor (PRR) proposed to bind conserved molecular structures on microbes (pathogen-associated molecular patterns, PAMPs, e.g., LPS) [34]; (2) as an active agent involved in the apoptotic cell cleaning process. Apoptotic cell-associated molecular patterns (ACAMPs) interfere with PRR and LPS-like structures, as revealed on apoptotic cells [35]. Increased values of BCL2 and p53 in septic cardiomyopathy were associated with increased mortality [36]. Thus, CD14 ligation with different ligands (PAMP or ACAMP) leads to an opposite response. CD14 binds LPS to generate proinflammatory responses or binds ACAMP for the clearance of apoptotic cells in a noninflammatory way [37,38]. It would be interesting to evaluate the level of CD14 in survivors of septic shock of comparable severity. This would prove the association between increased expression rates and poor outcome in septic shock. Various studies attempted to identify a role of complement, coagulation, and inflammation pathways in the pathophysiology of myocardial injury [39].

The microcirculatory disorders induced by the microorganism associated with destruction of myocardial fibers generate the release of oxygen species and may also contribute to the severity of septic myocarditis [40,41]. Myocardial biopsies cannot be obtained in critically ill patients. However, serum levels of mCD14 and sCD14 could be compared, considering their strong relationship. A comparison with different control groups, such as patients who died of trauma, could prove their specific significance in septic shock [41], and this is our goal for future work. Our current study was a quantitative retrospective study, but further investigations are needed to determine the minimum levels required of mCD14 and sCD14 to trigger death from septic cardiomyopathy [42].

A comparison with different control groups, i.e., patients who died of trauma, could prove of valuable significance in better understanding septic shock. 

This was a quantitative retrospective study, and further studies are needed to determine the minimum amount of LPS required to trigger death from septic cardiomyopathy.

## 5. Conclusions

The results of this study reveal a close positive association between the level of LPS related to sCD14 and the cellular expression of mCD14. By extrapolation, we suggest that a large amount of sCD14 detected in the myocardial tissue will activate the mCD14–TRL4–LBP–LPS complex and further induce an inadequate immune response, resulting in severe heart damage. A higher amount of LPS will induce more significant heart damage. Identification of the presence of mCD14 and sCD14 in the myocardium tissue can allow the indubitable diagnosis of septic cardiomyopathy.

## Figures and Tables

**Figure 1 diagnostics-12-00781-f001:**
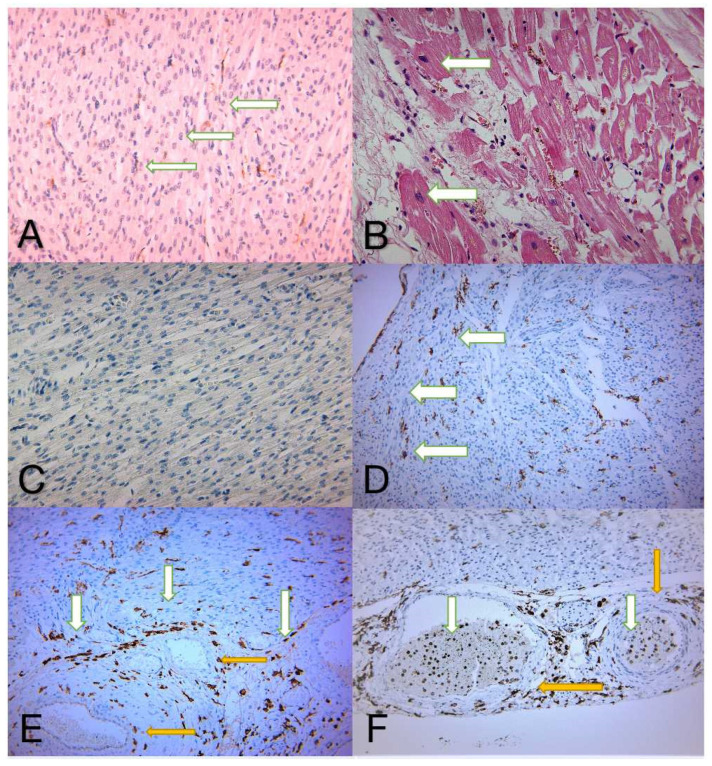
(**A**,**B**) H&E normal aspects of the control group, showing cardiac muscle fibers with interstitial edema and inflammatory infiltrate. (**A**) Cardiac muscle fibers with small nuclei located centrally—white arrows (magnification 100×). (**B**) The nuclei of the fibers are enlarged in volume, revealing cellular suffering—white arrows (magnification 200×). (**C**) The negative expression of CD14 in a specimen belonging to the control group. The causes of death in the control group were not related to heart disease (magnification 100×). (**D**) Immunohistochemical image of numerous myocardial fibers with inflammatory cells intercalated, with the membrane marked in brown, showing low CD14 expression (<10%)—white arrows (magnification 40×). (**E**) Immunohistochemical image of the myocardial cells, with numerous inflammatory cells intercalated and membranous immune expression of CD14 marked in brown and white arrows. Its expression is revealed mainly on the macrophage membrane surface, but also minimally distributed on the neutrophil surface of mCD14. No expression is seen in the blood vessels—yellow arrows (magnification 40×). (**F**) Immunohistochemical image of a blood vessel located intramuscularly—yellow arrows. In the vessel’s lumen there can be noticed granular areas of extracellular, soluble CD14 (brown color) along with monocytes and neutrophils (mCD14)—white arrows (magnifications 40×).

**Table 1 diagnostics-12-00781-t001:** Distribution of the study group according to the infection site.

Infection Site	No. of Patients	Percentage
Abdominal	3	10.3%
CNS	2	6.9%
Hematogenous	2	6.9%
Pulmonary	16	55.2%
Unknown	2	6.9%
Urinary	4	13.8%
Total	29	100.0%

**Table 2 diagnostics-12-00781-t002:** Demographical and epidemiological data in the study group and control group.

Characteristics	Study Group	Control Group	*p*-Value
Age (mean value)	0.3 years (±16.5)	4.2 months (±2.6)	
Gender F:M	18:11 cases	-	0.2652
Urban/Rural	16:13 cases	-	0.7103
Infection site origins	Lung Urinary tract	-	0.0001

**Table 3 diagnostics-12-00781-t003:** CD14 expression rate in the study group and the control group and age distribution.

Group	CD14 Expression Rate	
	<10%	>10%
Control (*n* = 10)	10	0
Study (*n* = 29)	10	19

**Table 4 diagnostics-12-00781-t004:** Age distribution according to CD14 expression in the study group.

	CD14 Expression Rate	
	<10%	>10%
Age, years (±SD)	53.70 (±20.8)	63.73 (±12.9)

**Table 5 diagnostics-12-00781-t005:** Comparison between different levels of membranous and soluble CD14 in the study group.

Intravascular CD14	Cellular CD14 Expression Rate	
	<10%	>10%
Increased	2	15
Decreased	8	4

**Table 6 diagnostics-12-00781-t006:** Levels of soluble CD14 in females and males of the study group.

Gender	Intravascular CD14	
	Decreased	Increased
Female	6	12
Male	6	5

**Table 7 diagnostics-12-00781-t007:** Age distribution according to different groups of soluble CD14.

	Intravascular CD14	
	Decreased	Increased
Age, years (±SD)	57.75 (±15.5)	62.05 (±17.4)

## Data Availability

Not applicable.

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
