# Peer review of "Assessing the Role of Lipopolysaccharide (LPS) Receptor (CD14) in Septic Cardiomyopathy: The Value of Immunohistochemical Diagnostics"

_diagnostics, 2022, doi:10.3390/diagnostics12040781_

Round 1

Reviewer 1 Report

In this paper the Authors have analyzed a series of 29 subjects dead by septic shock and evaluated CD14 immunohistochemical staining in autoptic heart tissues as marker of adverse evolution. This topic is of interest for the readership of  the journal, however the results are not completely convincing and exposed in a rather confusing way.

1) It is unclear the significance of the cutoff (10%) used to split the series; the Authors should better quantify CD14 immunostaining (for instance, number of positive cells for microscopic area)

2) I am not sure that newborns are the best control; the Authors may use cases of adult patients dead for non-cardiac diseases, or patients deceaded for non septic shock

3) Figures B and C should be zoomed; furthermore, CD31 or CD34 immunostaining should be performed in some cases to show at the best intravascular sCD14

4) English should be revised. Mistakes are disseminated in the text

Author Response

To Reviewer 1:

  1. Dividing the test subjects into two groups in correlation with the 10% expression value was imposed by the preliminary evaluation of data from the control group. All patients in the control group had less than 10% expression, thus the 10% value became the cut-off limit. This is a practice that other authors use in their studies for instance: Jeffery M. Klco,PhD, Shashikant Kulkarni,Friederike H. Kreisel, Tu-Dung T. Nguyen,, PhD, Anjum Hassan and John L. Frater.Usefulness of CD14 and Krüppel-like Factor 4, a Novel Monocyte Marker. Am J Clin Pathol 2011; 135:720-730 DOI: 10.1309/AJCPZ46PMMAWJROT
  2. We chose our control group to consist only out of newborns who had no cardiac diseases of malignancies to avoid data alteration in relation with CD14. It is also a well-known fact that newborns have an immune system that is yet to fully develop so the number of macrophages and polymorphonuclears present in myocardial tissue will thus be very small. Using as comparison a control group consisting of adults with different causes of death might have led to misinterpretations, but we also believe that this should be strongly considered for future studies.
  3. The microscopical images were modified in such a manner to better underline the results described. The use of CD31 an CD34 would of course better highlight the blood vessels but we consider that their histological characteristics in enough to recognize them. Nevertheless, to focus our attention on them we marked them with yellow arrows.

Reviewer 2 Report

A much better control group would be people of comparable age to the subjects. There is a lack of information as to why neonates were chosen. Results are presented in an incomprehensible manner. The data are incomplete or even wrong. Parts of the results should be carefully reworded. There is a lack of interpretation of the results based on a sequence of molecular events. 

Author Response

  1. We chose our control group to consist only out of newborns who had no cardiac diseases of malignancies to avoid data alteration in relation with CD14. It is also a well-known fact that newborns have an immune system that is yet to fully develop so the number of macrophages and polymorphonuclears present in myocardial tissue will thus be very small. Using as comparison a control group consisting of adults with different causes of death might have led to misinterpretations, but we also believe that this should be strongly considered for future studies.
  2. As requested, results were double-checked.
  3. We hope that at a careful review you will find adequate information about mechanism sequence of septic shock and CD14 involvement in the “Introduction” and “Discussions” chapters.

Reviewer 3 Report

Reil PM et evaluated in 29 patients who died of septic shock the values of membranous CD14 and soluble CD14 in the heart tissue. Correlating the expression with adverse patient evolution, the Authors found a high cellular expression of CD14 in the study group VS control group expression rate under 10% (p=0.0013).

The Authors concluded that  CD14  could represent a  valuable marker for septic cardiomyopathy, thus apoptosis of cardiomyocytes could be foreseen by its high value.

General comment

Thanks for allowing me to review this paper. The paper is interesting, but in the present form, I think I can't make a thoughtful judgment. There are many typos (see last line Table 1, first line Table 2, Legend of figures), and the first page of Discussion is missing.

However, I can make some comments:

1) The control group was done by newborns. I think It would be better to have a group of patients who died for unrelated septic reasons (such as acute cerebrovascular events) as the control group.

2) To look for a correlation between clinical features of septic patients and the levels of sCD14 and mCD14,  I suggest adding other clinical data, such as the requirement of vasopressor, the interval between the onset of septic shock and death, and so on.

2) The Authors took a cut-off of 10% of CD14 expression to categorize low-expression and high expression rates. Please add references for this cut-off value.

Author Response

  1. As requested, Tables 1 and 2 were modified.
  2. We chose our control group to consist only out of newborns who had no cardiac diseases of malignancies to avoid data alteration in relation with CD14. It is also a well-known fact that newborns have an immune system that is yet to fully develop so the number of macrophages and polymorphonuclears present in myocardial tissue will thus be very small. Using as comparison a control group consisting of adults with different causes of death might have led to misinterpretations, but we also believe that this should be strongly considered for future studies.
  3. This is a histopathological study by nature therefor clinical data may be scarce. We considered only some clinical data relevant for it. Being a pilot study, we think it could represent a relevant informational base for future studies that would exhaustively corroborate other clinical data for instance the time elapsed between the onset of septic shock and death.
  4. Dividing the test subjects into two groups in correlation with the 10% expression value was imposed by the preliminary evaluation of data from the control group. All patients in the control group had less than 10% expression, thus the 10% value became the cut-off limit. This is a practice that other authors use in their studies for instance: Jeffery M. Klco,PhD, Shashikant Kulkarni,Friederike H. Kreisel, Tu-Dung T. Nguyen,, PhD, Anjum Hassan and John L. Frater.Usefulness of CD14 and Krüppel-like Factor 4, a Novel Monocyte Marker. Am J Clin Pathol 2011; 135:720-730 DOI: 10.1309/AJCPZ46PMMAWJROT

Reviewer 4 Report

The study of Reil et al. describe a critical role of mCD14 in the pathogenesis of septic cardiomyopathy and the interaction of LPS related sCD14 and the cellular expression of mCD14. Technically the study was performed properly.  However, it is doubtful, if the control group was chosen correctly for the final statement. At least, the author state that a comparison with different control groups may be a next goal.

A closer look at which cells actually express mCD14 and a specific characterization of the positive cells would be interesting (in which compartment?).

A correlation with clinical data would improve the studies relevance.

Unfortunately, page 5 was unable to read and no figure legend is shown.

From which provider is the used antibody?

Table 2 is uncomplete (age? "es"?)

Author Response

  1. The types of inflammatory cells (macrophages and neutrophils) that showed CD14 expression were detailed highlighted here including the field of expression of the used antibody: membranous (mCD14) and extracellular granular (sCd14).
  2. This is a histopathological study by nature therefor clinical data may be scarce. We considered only some clinical data relevant for it. Being a pilot study, we think it could represent a relevant informational base for future studies that would exhaustively corroborate other clinical data with new cases of Sepsis-induced myocardial dysfunction.
  3. As requested, we named the producing company.
  4. As requested, Table 2 was modified.

Round 2

Reviewer 3 Report

I have no further comments